# Compressive Mechanical Properties of Larch Wood in Different Grain Orientations

**DOI:** 10.3390/polym14183771

**Published:** 2022-09-09

**Authors:** Jingcheng Sun, Rongjun Zhao, Yong Zhong, Yongping Chen

**Affiliations:** 1Institute of Ecological Conservation and Restoration, Chinese Academy of Forestry, Beijing 100091, China; 2Research Institute of Wood Industry, Chinese Academy of Forestry, Beijing 100091, China; 3Co-Innovation Center for Efficient Processing and Utilization of Forest Products, Nanjing Forestry University, Nanjing 210037, China

**Keywords:** compressive failure mechanism, larch wood, digital image correlation method (DIC), load–displacement curve, energy dissipation

## Abstract

As a green and low-carbon natural polymer material, wood has always been popular in engineering applications owing to its excellent physical and mechanical properties. In this study, compression tests in conjunction with in situ test methods (DIC method) were used to investigate the compression mechanism of wood samples in the longitudinal, radial, and tangential directions. The macroscopic failure modes, energy dissipation results, and variations in the strain field were analyzed. The results showed that the load–displacement curve in each grain orientation included three stages: an elasticity stage, yield stage, and strengthening stage. Both the compressive strength and elastic modulus in the longitudinal direction were significantly higher than those in the radial and tangential directions, but there was no significant difference between the radial and tangential directions. Specimens in the longitudinal direction mainly presented fiber buckling, fiber shear slippage, and fiber fracture failure; in radial directions mainly presented compression compaction of the fiber cells; and in the tangential directions presented buckling and shear failure of the laminar layers. The energy absorption in the longitudinal direction was better than in the other directions. The strain changed significantly in the loading direction in the elastic stage while the shear strain changed remarkably in the yield stage in each grain orientation. In this paper, the compression mechanical properties of larch wood in different grain orientations were studied to provide a reference for its safe application in engineering.

## 1. Introduction

As a green, environmentally friendly, low-carbon, and natural biopolymer material, wood has good physical and mechanical properties, and is an important building material [1]. Differing from the most widely used building materials, such as steel and concrete, wood is generally considered to be an anisotropic material. The direction parallel to the fiber is the longitudinal direction (*L*), the direction perpendicular to the growth ring is the radial direction (*R*), and the direction tangent to the growth ring is the tangential direction (*T*). The radial and tangential directions are collectively referred to as the transverse directions [2,3]. The mechanical evolution in different directions is key to guaranteeing reliability and safety during construction.

The compression resistance is one of the most important mechanical properties of wood when it is used as a structural material. Many studies have focused on the compression properties of wood in different directions. Previous studies showed that the compressive strength of wood under longitudinal compression was 8–10 times that in the transverse direction [4,5,6]. The characteristics of the stress–strain curves of wood under compression were also different. A great many constitutive models have been proposed to describe the compression properties of wood [7,8,9,10,11,12]. Additionally, in order to study the failure characteristics of wood, a large number of studies on the failure modes during longitudinal [13,14,15,16], radial [10,17], and tangential compression [18] have been conducted.

The failure mechanism of wood has also been widely researched, which explains the essence of the damage characteristics in different directions [19,20,21]. However, the studies on the anatomical structures in the related research were basically conducted using an optical microscope or scanning electron microscope to observe the sections of the specimens after loading, which had many limitations in revealing the failure mechanism of wood compression. To explore the compression failure mechanism of wood more effectively, it is necessary to analyze the whole failure process and the changes in the strain field using in situ test methods. The digital image correlation (DIC) method has been proved to be an effective test method and is widely used for wood composites and cross-laminated wood [22,23,24].

Thus, in order to explore the compression failure mechanisms of wood in three directions, the DIC method was used to study the damage characteristics and strain field changes of wood under longitudinal, radial, and tangential compression, respectively, and the macro-damage mechanism of wood under compression was also explored. The results of this paper will benefit the reasonable design and safe application of wood in wooden buildings.

## 2. Material and Methods

### 2.1. Materials

Larch (*Larix principis-rupprechtii* Mayr) comprises one of the three major artificial forests in China, with the characteristics of fast growth and strong resistance, meaning it is considered as an excellent building material. Therefore, larch was selected as the test material to be processed for the longitudinal, radial, and tangential compression samples. Larch samples were taken from Saihanba forest farm, Zhangjiakou City, Hebei province, with a diameter at breast height of 260 mm and a tree age of 31 years. In order to process the compression samples, straight and flawless air-dried wood strips with dimensions of 1000 mm (*L*) × 40 mm (*R*) × 40 mm (*T*) were sawed from larch logs. The strips were then processed into longitudinal, radial, and tangential specimens with sizes of 30 mm × 20 mm × 20 mm, respectively (Figure 1). Ten specimens were processed for each direction. The air-dried density and moisture contents of the samples were about 0.593 g/cm^3^ and 10.49%, respectively.

### 2.2. Test Method

A universal mechanical testing machine with a load capacity of 100 kN (Model: 5582, Instron Co., Ltd., Norwood, MA, USA) was applied to evaluate the compression properties of each specimen, as shown in Figure 2. The specimens were loaded with a loading rate of 1 mm/min until the load–displacement curve reached the hardening stage, referencing the Chinese standards GB/T 1935–2009 and GB/T 1939–2009. The compressive strength and elastic modulus were calculated according to Formulas (1) and (2):(1)$$\sigma =\frac{P}{bt}$$
(2)$$E=\frac{\Delta P\times l}{b\times t\times \Delta l}$$
where *σ* is the compressive proportional ultimate strength (MPa), *P* is the proportional ultimate load (N), *b* is the width of the sample (mm), *t* is the sample thickness (mm), *E* is the compression elastic modulus (MPa), Δ*P* is the load increment at the elastic stage of the stress–strain curve, l is the height of the sample (mm), and Δ*l* is the strain increment corresponding to Δ*P*.

A digital camera (AVT Prosilica GT4905, mono, 4896 × 3264 megapixels, 7.5 fps, Ruituotech Co., Ltd., Beijing, China) was applied to record all views to measure the deformation of the wood during the test. Some black spots measuring 0.007 inches in size were randomly coated with a soft sponge on the surface of each specimen before the testing. The image processing software Vic-2D v6 was used to find displacements of the black spots to measure the deformation behavior of the wood, such as the strain along the x-direction (exx), strain along the y-direction (eyy), and shear strain (exy).

## 3. Results and Discussion

### 3.1. Load–Displacement Curve

The load–displacement curves of wood compressed in the longitudinal (*L*), radial (*R*), and tangential directions (*T*) are shown in Figure 3. The load–displacement curves of *L*, *R* and *T* included three stages: (I) linear elastic stage, (II) yield stage and (III) strengthening stage.

Through the average load–displacement curves of the *L*, *R,* and *T* specimens in Figure 3, it can be seen that the load increased linearly with the increase in displacement in the first stage. The proportional limit loads were 25 kN, 1.70 kN, and 3.23 kN for *L*, *R,* and *T,* with corresponding displacement rates of 0.54 mm, 0.26 mm, and 0.51 mm, respectively. In the second stage, the load of *L* decreased obviously with the increase in displacement (Figure 3a). When the displacement increased to 21.50 mm, the load decreased from 25 kN to 7.46 kN. The load of *R* increased slowly with the increase in displacement (Figure 3b). When the displacement increased to 8.43 mm, the load increased from 1.70 kN to 3.89 kN. The load of *T* decreased slowly with the increase in displacement (Figure 3c). When the displacement increased to 19.71 mm, the load decreased to 3.17 kN. From the second stage of the load–displacement curves, the brittleness of the wood in longitudinal direction was greater than in the tangential and radial directions. In the last stage, the loads in all three directions increased sharply with the displacement.

Figure 4 shows the average compressive strength and elastic modulus of larch in three directions. The compressive strengths of *L*, *R*, and *T* were 44.90 MPa, 4.60 MPa, and 5.78 MPa, with corresponding elastic modulus values of 16.95 GPa, 1.13 GPa, and 0.61 GPa, respectively. The compressive strength of *L* was 9.77 and 7.78 times that of *R* and *T*, while the elastic modulus was 15.01 and 27.70 times that of *R* and *T*, respectively.

### 3.2. Failure Mode

In order to investigate the failure mode of the wood in three grain directions, the typical failure morphology of each stage corresponding to the load–displacement curve was observed (Figure 5). Figure 5a shows the failure mode of specimen *L*. In the linear elastic stage, namely stage I in Figure 3a, there was no obvious deformation in the specimen. In the stage II, the fibers at the top and bottom of the specimen were first crushed, then an irregular wrinkle band was generated in the middle of the specimen. Under the action of a compression load, shear slips occurred between the fibers near the wrinkle band, resulting in the separation of the fibers and the generation of cracks. With the expansion of the cracks along the fiber direction, the bearing capacity of the specimen decreased continuously. In stage III, the specimen broke under the load of compression, accompanied by fiber separation and delamination [25], and the compression direction turned from longitudinal to transverse eventually. However, the kink bands mentioned in previous studies were not observed [14,26].

Figure 5b shows the failure modes of specimen *R*. In the linear elastic stage, as shown in stage I in Figure 3b, there was no obvious deformation in the specimen. In stage II, plastic yield deformation occurred in the specimen. In this stage, the early wood in every growth ring was compacted first due to the action of compression perpendicular or approximately perpendicular to the orientation of the growth ring. In addition, the local area of the specimen deformed severely owing to the non-uniformity of the wood, resulting in an “accordion-like” deformation morphology [27]. Meanwhile, the resin in the specimen was extruded and overflowed. In stage III, the specimen was subject to shear failure under the action of the compression load.

Figure 5c shows the failure modes of the *T* specimen. In the linear elastic stage, the same as for the *L* and *R* specimens, there was no obvious deformation in the specimen, as shown in stage I in Figure 3b. In stage II, buckling of the laminated structure of the wood occurred in the middle of the specimen and cracks were generated between the layers, leading to fiber slippage and slip separation between some of the fibers [18,28]. In stage III, the fibers fell down owing to the action of compression, and the specimen switched to radial compression.

In summary, it was found that the failure modes of the wood in the longitudinal direction were mainly fiber buckling, shear failure, and fiber breakage; the failure mode of the wood in the radial direction was compression deformation; and the failure modes of the wood in the tangential direction were growth ring buckling and fiber shear slip failure.

### 3.3. Energy Absorption Characteristics of Wood under Compression Load

The energy absorption of the wood can be expressed by the absorbed deformation energy per unit volume and calculated by the area of the stress–strain curve and coordinate axis [29]. The energy absorption–strain curve of the wood under compression load in different directions is presented in Figure 6. It was obvious that the energy absorption capacity of *L* was much greater than for *R* and *T*, and that of *T* was slightly greater than that of *R* when the strain was less than 0.28. This was because the compressive strength of *L* was significantly higher than in the other two directions, and *T* was slightly higher than that of *R*. The larger compressive strength had greater elastic potential energy, so it had greater energy to resist the external load.

When the strain was 0.70 or more, the energy dissipation characteristic of *R* exceeded that of *L*. According to the failure mode and load–displacement curve mentioned above, the shear failure of *L* caused the appearance of strain softening in the load–displacement curve, which resulted in a downward trend in the energy consumption of the unit volume. In the case of the *R* specimen, the compression deformation of the wood cells made the load–displacement curve rise slowly, resulting in a rise tendency for energy consumption of the unit volume. Therefore, the energy consumption of *R* exceeded that of *L* when the strain was about 0.70. Similarly, the energy consumption of *R* exceeded that of *T* when the strain was about 0.28.

The energy absorption efficiency is another important parameter used to evaluate the energy absorption of wood, which can be used evaluate the working state of wood under compression load. The energy absorption efficiency of the specimens was calculated using the ratio of the energy absorption of the wood for a practical application to that under ideal conditions. The calculation formula is (3) [30]:(3)$$\eta =\frac{\int_{0}^{\varepsilon_{m}}\sigma d\varepsilon }{\sigma_{\max}\varepsilon }$$
where *η* is the energy absorption efficiency (%), *σ* is the stress (MPa), *ε* is the strain, *σ*_max_ is the maximum displacement from 0 to *ε*_m_ (MPa), and $\varepsilon_{m}$ is the strain at the end of the loading.

The energy absorption efficiency levels of the wood compression in three directions were compared, as illustrated in Figure 7. It can be seen that the energy absorption efficiency of *L* was larger than for *R* and *T*. The energy absorption efficiency of *L* was 1.91 times that of *R* and 3.64 times that of *T*. According to the failure modes of the wood in the different directions, the energy dissipation of *L* mainly depended on the plastic slippage and the buckling of the fibers; the energy dissipation of *R* was mainly determined by the cell wall buckling and cell cavity collapse; the energy dissipation of *T* was dominated by the growth ring buckling and interlayer separation. The compressive strength and elastic modulus of *L* were larger than those of *R* and *T* (Figure 4). The larger elastic modulus and compressive strength of *L* caused greater elastic deformation energy, so the energy absorption efficiency of *L* was higher than for *R* and *T*.

### 3.4. Strain Field Analysis

Through the analysis of the failure modes and energy dissipation of the wood in different directions, the differences could be judged. In order to further analyze the reasons for the differences, the DIC method was used to monitor the surface strain fields of the samples. The strain field distributions in the elastic stage (stage I in Figure 3) and the yield stage (stage II in Figure 3) were analyzed, as shown in Figure 8 and Figure 9, respectively. In the elastic stage, the compression load in each direction selected was about 1.60 kN. In the yield stage, the selected compression loads of *L*, *R*, and *T* were about 25.73 kN, 2.33 kN, and 4.10 kN, respectively.

In the elastic stage, during the compression of *L* (Figure 8a), the change in exx was not obvious, and the value of the strain on the specimen surface was less than 0.001; the maximum value of eyy was approximately −0.002, and the strain concentration area was at the top of the specimen; the maximum value of exy was approximately −0.001, and the area of the strain concentration was the same as eyy. For the *R* specimen (Figure 8b), the maximum values of exx and exy were approximately 0.003 and −0.015, respectively, and their maximum strain concentration areas were both located at the top of the specimens; the maximum value of exy was approximately 0.004, located at the top of the specimen. It is worth noting that there were two regions with opposite directions of exy at the bottom and right sides of the specimen, which indicated that the specimen would undergo shear deformation in the opposite directions. For the *T* specimen (Figure 8c), the maximum value of exx was approximately 0.006, and the maximum strain concentration area appeared in the lower right corner of the specimen; the maximum value of eyy was approximately −0.016, and the maximum strain concentration area was located at the top of the specimen; the maximum value of exy was approximately −0.009, which appeared in the lower right corner of the specimen. Through the analysis of the strain field distributions in the elastic stage, the strain of *L* was significantly less than those of *R* and *T* when the load was about 1.60 kN. The elastic modulus of *L* was much greater than those of *R* and *T* (Figure 4). Therefore, the deformation in the longitudinal direction was less than those in the radial and tangential directions under the same load within the elastic range.

In the yield stage, the maximum values of exx, exy, and exy were approximately 0.004, −0.032. and 0.013, respectively, and all the strain concentration areas appeared at the top and bottom of the specimens during the compression of *L* (Figure 9a). As shown in Figure 9b, the strain concentration of exx in the *R* specimen contained two areas located in the top and lower left corners of the samples, respectively. The maximum strain values in the top and lower left corners of the specimens were about 0.007 and 0.005, respectively. The strain concentration of eyy also contained two areas located at the top and middle of the samples, respectively. The maximum strain at the top of the sample was about −0.061 and that at the middle of the sample was about 0.035. The strain concentration of exy contained two areas equally in the middle and lower parts of the specimen, and the maximum values of strains in these two regions were approximately 0.015 and −0.007, respectively. For the *T* specimen (Figure 9c), the strain distribution of exx was banded on the whole sample surface, and the maximum value of the strain was about 0.019, which appeared in the lower right corner of the specimen. The strain concentration area of eyy appeared at the top of the sample, and the maximum strain was about −0.084. The strain concentration of exy contained two areas that were basically symmetrically distributed, and their maximum strains were both −0.046. Through the analysis of the strain field distribution in the different directions, it was found that the strain concentration areas were related to the deformation modes of the specimen in each direction. During the compression of *L*, the fibers at the top and bottom of the specimen buckled and deformed, so the strain concentration appeared at the top and bottom; when compression occurred in the *R* specimens, “accordion-like” deformation occurred in the specimen, so there were two strain concentrations in the middle and lower parts of the specimen; for *T* specimens, buckling deformation and shear failure occurred, and as a result the strain concentration in the shear strain was very obvious.

## 4. Conclusions

In order to reveal the effect of the grain direction on the compressive failure mechanism of the wood, larch wood specimens in the longitudinal, radial, and tangential directions were tested. Through the analysis of the load–displacement curve, failure characteristics, energy dissipation performance, and strain field results of the specimens, the main conclusions were drawn as follows:(1)The load–displacement curves of larch wood in each grain direction contained elasticity, failure, and strengthening stages. The compressive strength in the longitudinal direction was 9.77 and 7.78 times those in the radial and tangential directions, and the elastic modulus in longitudinal direction was 6.96 and 10.94 times of those in the radial and tangential direction, respectively;(2)The failure modes of the compression in the longitudinal direction were mainly fiber buckling, fiber separation, and fiber fracture; the failure modes in the radial direction were mainly cellular compactness and accordion-like deformation; the failure modes in tangential direction were mainly growth ring buckling deformation and shear slippage between growth rings;(3)The energy absorption in the longitudinal direction was higher than in the radial and tangential directions. The energy absorption efficiency in the longitudinal direction was 1.91 and 3.64 times those in the radial and tangential directions, respectively;(4)The strain change in the longitudinal direction was smaller than those in the radial and tangential directions. In addition, the shear strain increased significantly in the yield stage.

## Figures and Tables

**Figure 1 polymers-14-03771-f001:**
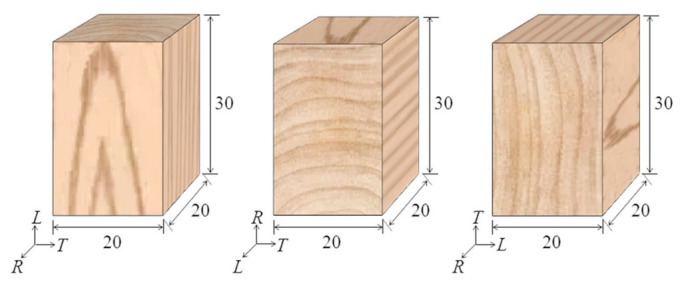
Schematic diagram of the specimens.

**Figure 2 polymers-14-03771-f002:**
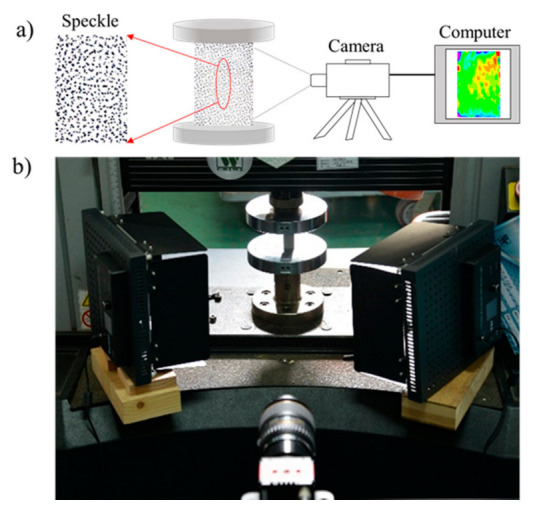
Schematic drawing of compression testing: (**a**) speckle on the specimen’s surface; (**b**) test schematic.

**Figure 3 polymers-14-03771-f003:**
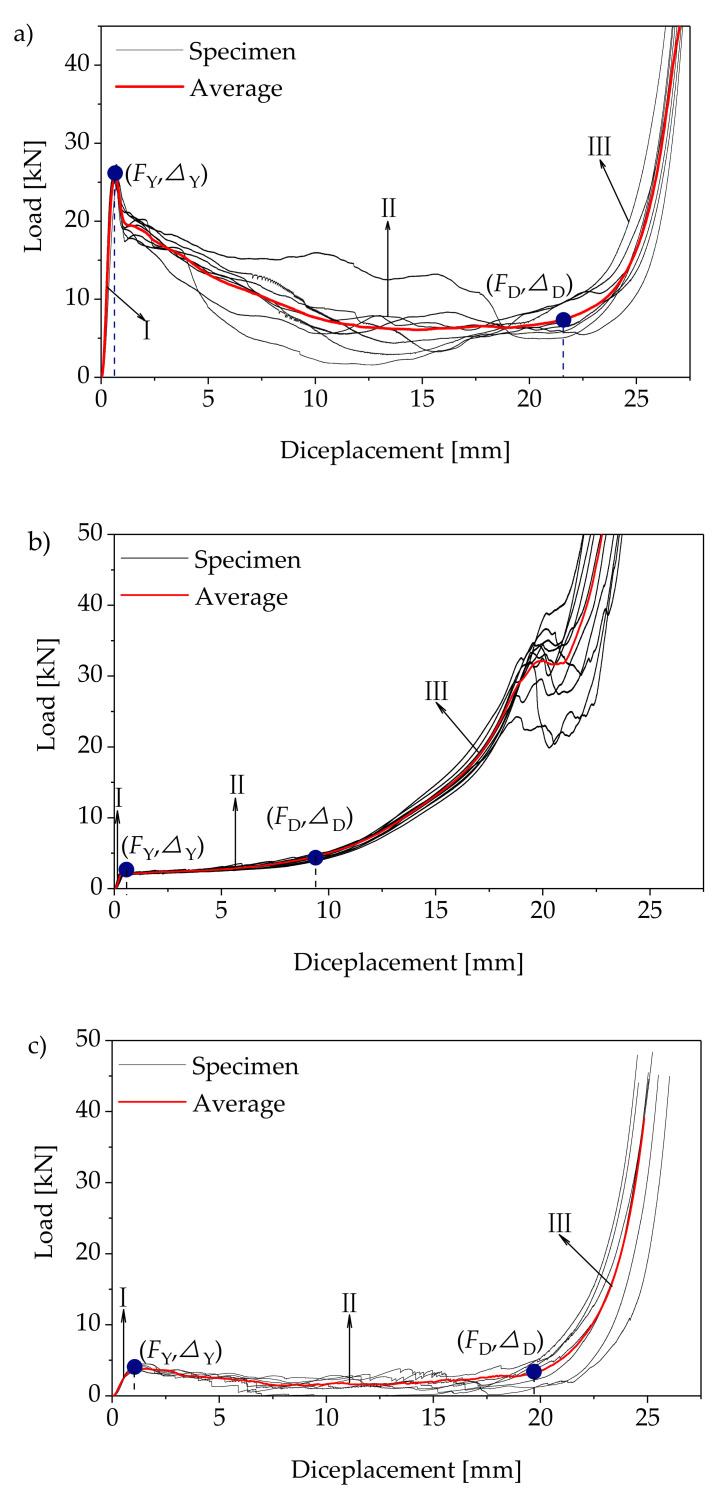
Stress–strain curves in different directions: (**a**) longitudinal direction; (**b**) radial direction; (**c**) tangential direction. Note: *F*_Y_ is the proportional limit load (kN); ∆_Y_ is the displacement corresponding to *F*_Y_; *F*_D_ is the dense point load (kN); ∆_Y_ is the displacement corresponding to *F*_D_.

**Figure 4 polymers-14-03771-f004:**
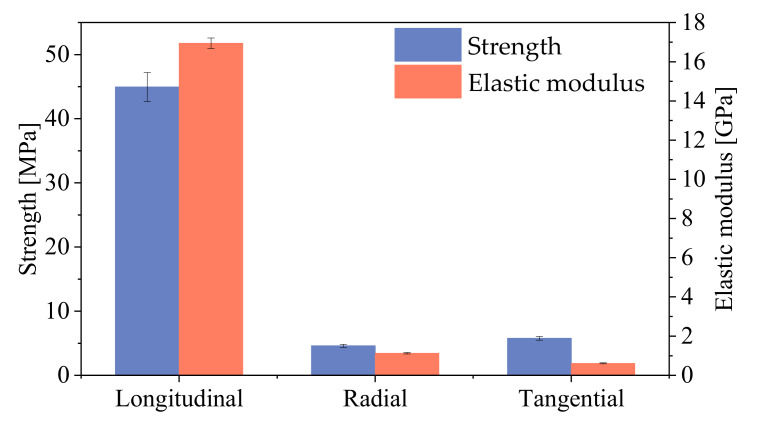
The compressive strength and elastic modulus of larch samples in different directions.

**Figure 5 polymers-14-03771-f005:**
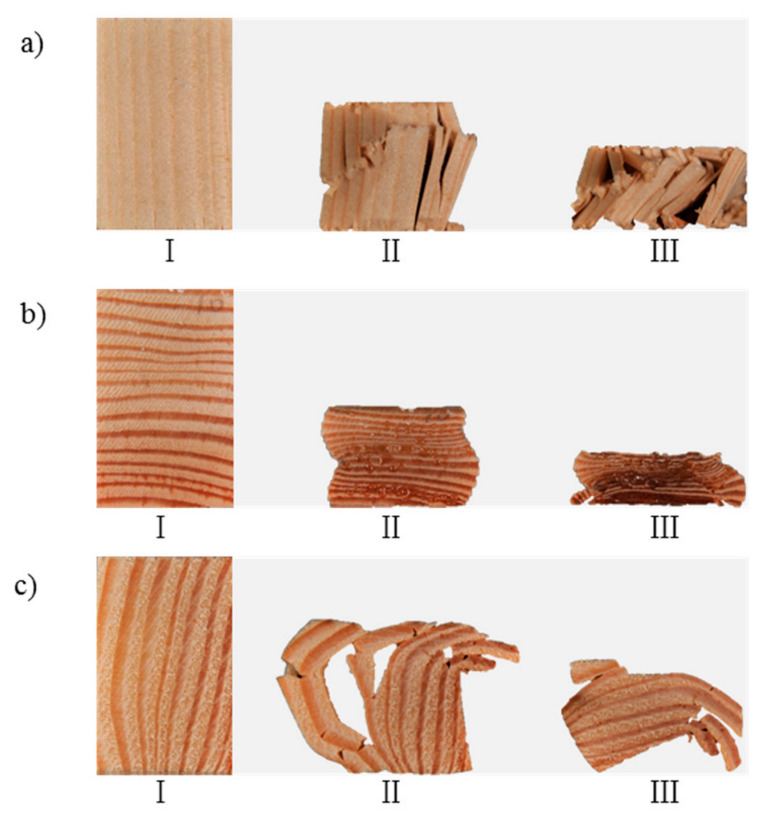
Failure modes in different directions: (**a**) longitudinal direction; (**b**) radial direction; (**c**) tangential direction. Note: I, II, and III indicate different stages during the failure process.

**Figure 6 polymers-14-03771-f006:**
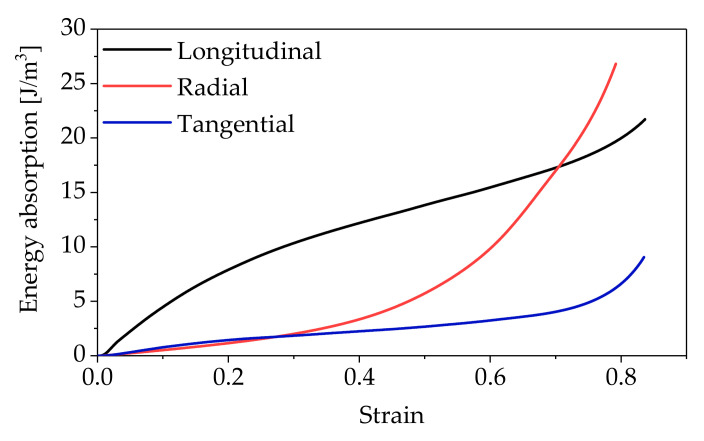
The energy dissipation–strain curves.

**Figure 7 polymers-14-03771-f007:**
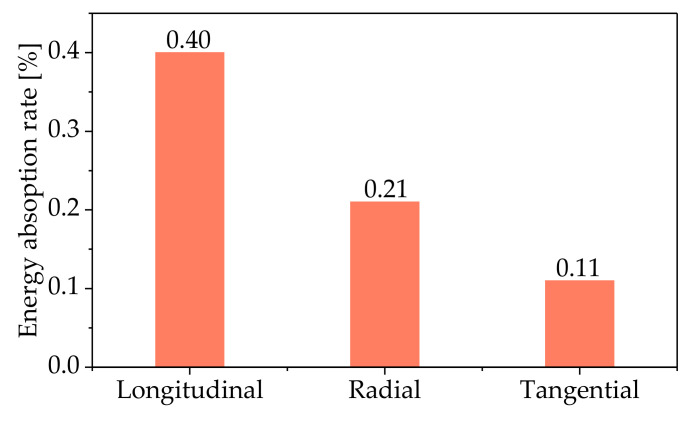
The energy absorption efficiency of the wood in different directions.

**Figure 8 polymers-14-03771-f008:**
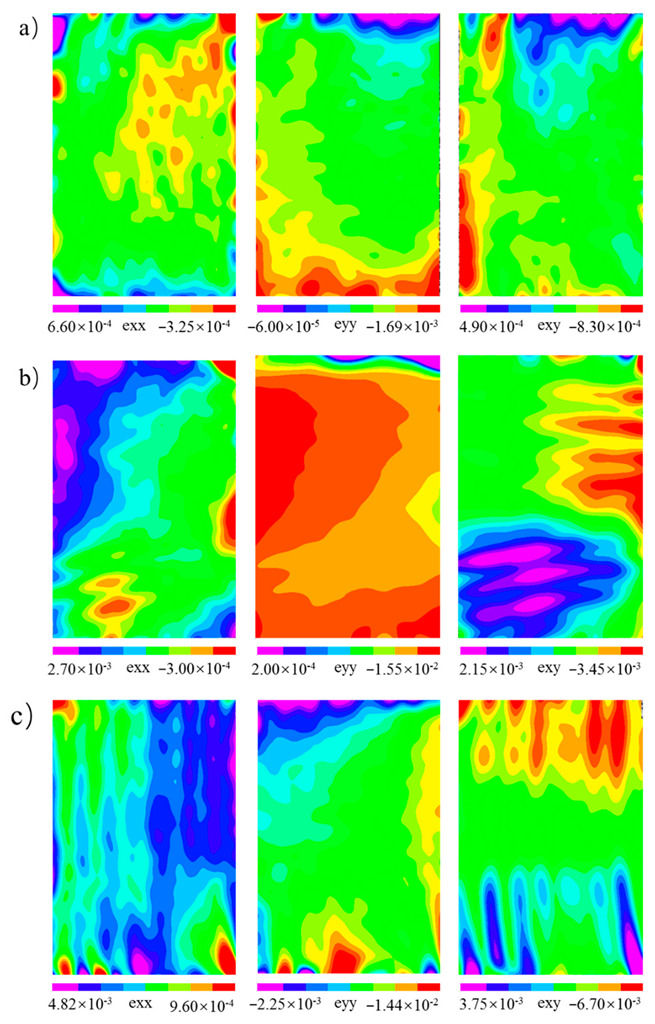
The distribution of the strain field in the elastic stage (*F* = 1.60 kN): (**a**) longitudinal direction; (**b**) radial direction; (**c**) tangential direction. Note: In the figures, exx represents the strain perpendicular to the load direction, eyy represents the strain parallel to the load direction, and exy represents the shear strain.

**Figure 9 polymers-14-03771-f009:**
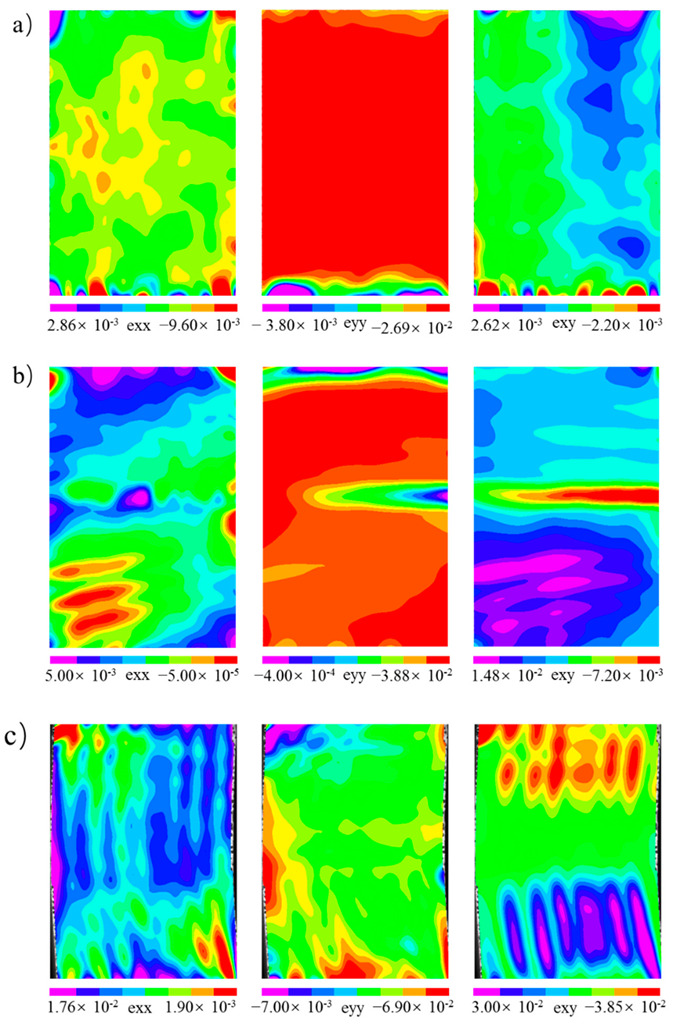
The distribution of the strain field in the yield stage: (**a**) longitudinal direction (*F* = 25.73 kN); (**b**) radial direction (*F* = 2.33 kN); (**c**) tangential direction (*F* = 4.10 kN).

## Data Availability

The data presented in this study are available on request from the corresponding author.

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
