# Peer review of "Compressive Mechanical Properties of Larch Wood in Different Grain Orientations"

_polymers, 2022, doi:10.3390/polym14183771_

Round 1
Reviewer 1 Report
The article Compressive Mechanical Properties of Larch Wood in Different Grain Orientations is well and clearly written, and the results are to be expected for wood as a material, since it has long been known that it has different mechanical properties in longitudinal, radial, and tangential directions. The information is not new. Only a new technique has been used. Microscopy of the specimens would be of interest.
Reviewer 2 Report
Comments (not suggestions) for authors
The compression strength in the three axis has been extensively studied; however, the authors suggest that DIC has not been coupled with traditional compression strength studies. The authors need to add this novelty to the abstract so that readers will know that this is new work.
After reading the DIC method in the methods section, it seems like this method is not completely described to the reader.
How did you decide to stop the stress strain curve when it reaches phase III? Because the load will increase rapidly and you can break the load cell.
Section 3.2 - the title should be capitalized.
The pitting shapes in the radial and tangential face can be quite different for some species. Since you did not see differences in radial and tangential compression, it probably means the shapes are more similar than other species.
You say “Meanwhile, the resin in the specimen was extruded and overflowed…” Do you mean adhesive or do you mean rosin from the tree?
According to figure 5 the compressive deformation is quite different between the radial and tangential direction. The radial looks weaker.
The rheology of wood / creep is not discussed. The compressive creep in the three different directions will be dependent on the deformation of the lignin / noncrystalline matrix between the microfibrils - Forest Products Journal (2022) 72 (2): 116–119. The microfibrils will be at some angle to the grain direction and loading orientation. This reference could be added to help provide some insight into compressive creep for larch.
Reviewer 3 Report
The article presented for review concerns the technical evaluation of a selected wood species. The authors decided to evaluate the mechanical properties in a compression test of larch wood. Tests were performed for all three anatomical directions of the wood. During the tests, the authors supported themselves with the digital image correlation (DIC) method. The article is correctly written and contains all the essential elements for such a scientific paper. The discussion of the results is well documented. The data presented are precise and well prepared. The authors refer to publications by other authors in their analysis.
Some minor comments:
- The abstract is a little too extended - it should include the aim, a brief description of the research and present the main results.
- There is no precise aim marked at the end of the introduction.
- There is no information on why the authors chose larch.
- There is no information on how the authors applied the speckles with a diameter of 0.007 inches.
- What hardware and software were used for the DIC? This information should be included in the paper.
- Formulas 1 and 2 are basic, and there is no need to include them. However, if the authors feel that they help in understanding the content, then they can stay.
Reviewer 4 Report
Very interesting and original paper. Here are a few comments to be considered to eventuelly improve further the manuscript.
Test samples: originating from only one tree?
Test sample dimension in longitudinal direction: the ratio longitudinal/transverse directions seems rather low, at least according to ASTM standards
Page 3: replace b is the height by l is the height
Page 4: longitudinal elastic modulus of dry larch wood of a density above 0.5 g/cm3 is normally above 10 GPa; could the small dimension ratio longitudinal/transverse directions be an explanation? What it is called the press plateau effect on strain is greater if the dimension ratio is low. May be the result of Fig. 8 could help to address this question? For future research, it would be interesting to analyse the dimension ratio in the three structural directions.
Round 2
Reviewer 1 Report
As the author stated in their next study, DIC method will be combined with anatomical structure analysis to further study the wood failure mechanism. This will be interesting.